

# The effect of long-term volleyball training on the level of somatic parameters of female volleyball players in various age categories

Petr Kutáč[1], David Zahradnik[1], Miroslav Krajcigr[1] and Václav Bunc[2]

[1] Human Motion Diagnostics Center, University of Ostrava, Ostrava, Czech Republic
[2] Charles University Prague, Prague, Czech Republic

Corresponding author
Petr Kutáč, petr.kutac@osu.cz

## ABSTRACT

Volleyball is an exceedingly popular physical activity in the adolescent population, especially with females. The study objective was to assess the effect of volleyball training and natural ontogenetic development on the somatic parameters of adolescent girls. The study was implemented in a group of 130 female volleyball players (aged $12.3 \pm 0.5 - 18.1 \pm 0.6$ years) along with 283 females from the general population (aged $12.3 \pm 0.5 - 18.2 \pm 0.5$ years). The measured parameters included: body height (cm), body mass (kg), body fat (kg, %), visceral fat ($cm^2$), body water (l), fat free mass (kg) and skeletal muscle mass (kg, %). Starting at the age of 13, the volleyball players had significantly lower body fat ratio and visceral fat values than those in the general population ($p < 0.001$ in body fat % and $p < 0.01$ in visceral fat). In volleyball players, the mean body fat (%) values were $17.7 \pm 6.6$ in 12-year-old players, $16.7 \pm 4.9$ in 13-year-old players, $18.5 \pm 3.9$ in 16-year-old players, and $19.3 \pm 3.1$ in 18-year-old players. In the general population, the mean body fat (%) values were $19.6 \pm 6.3$ in 12-year-old girls, $21.7 \pm 6.4$ in 13-year-old girls, $23.4 \pm 6.1$ in 16-year-old girls, and $25.8 \pm 7.0$ in 18-year-old girls. The visceral fat ($cm^2$) mean values were $36.4 \pm 19.3$ in 12-year-old players, $39.2 \pm 16.3$ in 13-year-old players, $45.7 \pm 14.7$ in 16-year-old players, and $47.2 \pm 12.4$ in 18-year-old players. In the general population, the mean visceral fat ($cm^2$) values were $41.4 \pm 21.1$ in 12-year-old girls, $48.4 \pm 21.5$ in 13-year-old girls, $58.0 \pm 24.7$ in 16-year-old girls, and $69.1 \pm 43.7$ in 18-year-old girls. In volleyball players, lower body fat ratio corresponded with a higher skeletal muscle mass ratio. The differences found in skeletal muscle mass ratio were also significant starting at the age of 13 ($p < 0.001$). The mean skeletal muscle mass (%) values were $44.1 \pm 3.4$ in 12-year-old volleyball players, $45.4 \pm 2.5$ in 13-year-old players, $45.0 \pm 2.2$ in 16-year-old players, and $44.7 \pm 1.8$ in 18-year-old players. In the general population, the mean skeletal muscle mass (%) values were $42.8 \pm 3.2$ in 12-year-old girls, $42. \pm 4.1$ in 13-year-old girls, $41.9 \pm 3.3$ in 16-year-old girls, and $40.6 \pm 3.7$ in 18-year-old girls. Differences in body composition between the individual age groups were similar between the volleyball players and girls in the general population. The results indicate that regular volleyball training influences the body composition of young females however the development of body composition parameters is subject to their ontogenetic development.

## INTRODUCTION

Volleyball is currently considered to be a dynamic game, during which low intensity and high intensity movements alternate. The high intensity movements include jumps, shuffles and rapid changes in direction (*Calleja-Gonzalez et al., 2019*). The offensive and defensive skills in volleyball are characterized as double-leg take-off and double-leg or single-leg landings (*Tillman et al., 2004a*; *Lobietti et al., 2010*; *Zahradnik et al., 2017*). A study by *Tillman et al. (2004b)* showed 1,087 jump-landings in two matches among four NCAA Division IA female volleyball teams. A study by *Lobietti et al. (2010)* reported 2,022/2,273 jump-landings for male/female players in six matches in each category of the Italian league. Similarly, *Zahradnik et al. (2017)* showed 992/1,375 jump-landings during three matches in elite volleyball teams in the Czech Republic. The total intensity of the movements in volleyball can be measured at 6 METs (*Scribbans et al., 2015*), which is considered to be vigorous (*Ainsworth et al., 2011*). Therefore, a corresponding level of physical conditioning is required to effectively cope with the load in the long term. This physical conditioning is achieved with regular physical activity, which is performed during the training process. The training process in volleyball starts in the period of the second childhood (Infans II), which is related to the age categories in the system of competitions in the Czech Republic organised by the Czech Volleyball Federation. The competitions organised within the Mini-volleyball in Colours project are for children from the age of nine. The long-term preparation should influence the development of specific physical skills as well as the somatic parameters of the volleyball players. In particular, on their body composition as it is a result of the level of adaptation of the organism to the load within the conditional preparation (*Tota et al., 2019*). This adaptation is manifested not only in the motor performance of the athlete, but also on their physical fitness and health (*Malá et al., 2015*).

Regular and adequate training since childhood should not only lead to the development of fitness and motor performance, but also to the development of a series of personal traits of the individual (*Kahlin et al., 2016*; *Joyner & Loprinzi, 2018*). Many of those personal traits are considered to be the predictor for adherence to physical activity (*Annesi, 2004*). These traits include: the ability to exert effort, determination, diligence, inclination to continue in the activity and to fulfil the task (*Annesi, 2004*). A regular physical activity might thus become a part of the lifestyle of these individuals. The inclusion of an adequate physical activity in one's daily routine is very important with respect to the individual's health, which may also be assessed through body composition assessment. The contribution of the physical activity to health and optimisation of body composition parameters has been documented in many studies (*Haskell et al., 2007*; *Lazaar et al., 2007*; *Nelson et al., 2007*; *Roriz et al., 2016*; *Bunc, 2018*). The importance of adherence to physical activity since childhood is currently very important as many studies show that the amount of childrens physical activity that meets the minimum level of physical activity for health

benefits is decreasing (*Liu et al., 2013*; *Sigmund et al., 2015*). Between 2011 and 2016, a stable insufficient level of physical activity is stated, with the highest occurrence being reported in the wealthiest countries (*Guthold et al., 2018*). In the Czech Republic, the documented physical routine in children dropped by about 30% in the last two decades. The decrease in physical activity of children is also inversely related to their age (as their age increases, their spontaneous physical activity decreases) (*Bunc, 2018*). A question arises as to whether or not using volleyball as the only physical activity can lead to an adjustment of somatic parameters (such as body mass), increase in fitness and thus provide considerable health benefits. The form and popularity of a physical activity are fundamental to its habitualization in both child and adolescent populations (*Ennis, 2017*; *Laroche, Girard & Lemoyne, 2019*). Volleyball is one of the frequently used physical activities in those age categories (*Liu et al., 2013*; *Glinkowska & Glinkowski, 2018*). Data from the Czech Volleyball Association indicates it is a popular sport in the Czech Republic where the number of registered children increased by 43.7 % between 2008 and 2018.

There are several studies that have researched the body composition of female volleyball players (*Nikolaidis, 2013*; *Visnes & Bahr, 2013*; *Ćopić et al., 2014*; *Paz et al., 2017*; *Valente-Dos-Santos et al., 2018*). These studies have analysed differences between volleyball and other sports, between female volleyball players and untrained individuals, body composition in relation to the game position of the players, the effect of training on body composition and the occurrence of injuries, or the effect of body composition on the physical ability of players. However, there is a lack of data on the development of body composition parameters in female volleyball players through their ontogenetic development, which would also answer the question about whether or not regular volleyball training may influence the natural ontogenetic development of the parameters. The answers to these questions should guide volleyball trainers to better interpret the values of somatic body composition parameters and avoid poor conclusions that might be reflected in inappropriate training plans.

The study objective was to assess the effect of volleyball training and natural ontogenetic development on the somatic parameters of adolescent girls.

Two hypotheses were formulated for verification in the study.

H1: Volleyball training of adolescent girls significantly influences their body composition.

H2: The trend of long-term changes in the selected body composition parameters is influenced by the natural ontogenetic development of young females.

## METHODS

### Participants

The study included a total of 413 participants (130 female volleyball players and 283 girls from the control group - general population). The detailed characteristics of the number and age of the participants is presented in Table 1. Participants had no medical difficulties and were not currently taking any medication or food supplements. Only those who were not menstruating were measured. Participants provided the information prior to the

**Table 1  Numbers and age of participants.**

| Age group | Volleyball group (VG) | | Control group (CG) | | Difference |
|---|---|---|---|---|---|
| | n | M ± SD (years) | n | M ± SD (years) | years |
| G1 | 27 | 12.32 ± 0.50 | 49 | 12.25 ± 0.49 | 0.07$^{N+}$ |
| G2 | 42 | 13.75 ± 0.46 | 76 | 13.85 ± 0.51 | 0.17$^{N+}$ |
| G3 | 31 | 16.08 ± 0.62 | 77 | 16.03 ± 0.48 | 0.05$^{N+}$ |
| G4 | 30 | 18.06 ± 0.60 | 81 | 18.15 ± 0.53 | 0.09$^{N+}$ |

Notes.

n, frequency; M, mean; SD, standard deviation; N, no significant.

$^{+}d \leq 0.2$.

measurement. They participated voluntarily and they were informed of the course of the study in advance. All participants signed an informed consent prior to participation in the study (the consent was signed by legal guardians for participants who were below the age of 18). The study was approved by the Ethical Committee of the Faculty of Education at the University of Ostrava (PdF OU č. 18/2018) and it is in compliance with the Helsinki Declaration.

The age division of volleyball players is based on the competition rules of the Czech Volleyball Association. Group VG1 is the category of Younger Pupils (12 years old and younger), VG2 is the category of Older Pupils (13–14 years old), VG3 is the category of Cadets (15–16 years old) and VG4 is the category of Juniors (17–19 years old). To be included in the study, the player had to be registered in the list of a given age category in a volleyball club. All the players played the highest level of competitions in the given age category in the Czech Republic. They were all players from teams in the Moravian-Silesian Region. The total number of female volleyball players was 810 based on an analysis of registered volleyball teams competing at the highest level in the given age category in the Czech Republic. The number of the monitored players represented 16% of all players in the Czech Republic. The control group included girls based on an intentional selection to avoid significant differences in the age between VG and CG.

Physical activity of the participants:

The girls in the general population had compulsory physical education twice a week. In their free time, they did not pursue any other regularly organised physical activity.

The volleyball players also had compulsory physical education twice a week. In their free time, they only pursued volleyball activities. The detailed schedule of volleyball activities during the season of competitions is presented in Table 2. The frequency and duration of matches is based on the system of the volleyball competition organisation. The data on the volume and frequency of training were obtained from responsible trainers.

## Procedures

The measurements of VG and CG were done in the morning during the autumn (September–October) in 2018. The volleyball players were measured in the Human Motion Diagnostic Centre laboratory and CG girls were measured at schools. The measurements and data evaluation were performed by the authors. All the measurements were executed with adherence to the principles of measurement using the bioelectric impedance method
**Table 2** Frequency and volume of weekly physical activity of the female volleyball players.

| Age group | Training / week | | Match / week | | Total time (min) |
|---|---|---|---|---|---|
| | Frequency | Time (min) | Frequency | Time (min) | |
| VG1 | 3 | 300 | 1 | 60 | 360 |
| VG2 | 4 | 450 | 1 | 60 | 510 |
| VG3 | 4 | 480 | 2 | 100 | 580 |
| VG4 | 4 | 480 | 2 | 100 | 580 |

**Notes.**
VG, volleyball group.

(BIA) (*Kyle et al., 2004*). Body height (BH) was measured using Stadiometer InBody BSM 370 (Biospace, South Korea), body mass (BM) and body composition were measured by the InBody 770 analyser (Biospace, South Korea). It is a tetrapolar multi-frequency bioimpedance analyser that uses the frequency of 1000 kHz for measurements and that is also a scale. The body composition parameters measured were body fat (BF) and visceral fat (VFA) expressed as area ($cm^2$), total body water (TBW), fat free mass (FFM) and skeletal muscle mass (SMM).

## Statistical analysis

The normality of data division was checked by the Shapiro–Wilk test. To assess the statistical significance of the differences in the means between the somatic parameters of the volleyball players and the girls in general population, a parametric independent $t$-test was used. The level of statistical significance for all the used tests was set at $\alpha = 0.05$. Practical significance was assessed using the effect of size (ES) by Cohen (Cohen's $d$). The $d$ value at the level of 0.2 indicates a minor difference, 0.5 an intermediate difference and 0.8 a major difference (*Cohen, 1988*). We considered the value of Cohen's $d \geq 0.5$ to be practically significant. We used Cohen's effect of size because it enables the assessment of the size of the difference between groups independent of the sample size. To verify whether or not we could consider our control group of girls to be general population, we compared the values of their basic anthropometric parameters of BH, BM and the calculated BMI with the values of the 6th Nation-wide Anthropological Survey of Children and Adolescents (*Kobzová et al., 2004*). For comparison, we used the normalization index (N $_i$).

Ni calculation: Ni $= \frac{MCG-M}{SD}$ (MCG –Mean control group, M –Mean 6th NAS, SD –standard deviation 6th NAS)

The Ni value in the range of $\pm 0.75$ SD shows an average development of the indicator, in the range from $\pm 0.76$ to 1.5 SD a below average (above average) development of the indicator, and the value above $\pm 1.5$ SD means a highly below average (above average) development.

To verify the accuracy of the measured body composition parameters, we used the measurement standardisation for the InBody 770 analyser in our laboratory. For that purpose, our laboratory uses the calculation of the typical error of measurement (TEM) and intraclass correlation (ICC) from three repeated consecutive measurements according to *Hopkins (2000)*. The measurements were done on 63 participants (24 female, 39 male) in the mean age of $21.96 \pm 2.57$ years. The TEM and ICC values are presented in Table 3.

**Table 3** Typical error of body composition parameters.

| Parameters | TEM (95% CI) | ICC |
|---|---|---|
| BM (kg) | 0.02 (0.02, 0.02) | 1.00 |
| TBW (l) | 0.12 (0.10, 0.13) | 1.00 |
| TBW (%) | 0.22 (0.19, 0.25) | 1.00 |
| BF (kg) | 0.24 (0.21, 0.27) | 1.00 |
| BF (%) | 0.29 (0.25, 0.33) | 1.00 |
| VFA (cm$^2$) | 0.92 (0.51, 1.05) | 1.00 |
| FFM (kg) | 0.23 (0.21, 0.27) | 1.00 |
| SMM (kg) | 0.14 (0.12, 0.16) | 1.00 |
| SMM$_p$ (%) | 0.16 (0.14, 0.19) | 1.00 |

**Notes.**
BM, body mass; TBW, total body water; BF, body fat; VFA, visceral fat area; FFM, fat free mass; SMM, skeletal muscle mass; TEM, typical error of measurement; 95% CI, confidence interval; ICC, interclass correlation.

The statistical processing of the results was performed using IBM SPSS Statistics (Version 21; IBM, Armonk, NY, USA).

## RESULTS

The results for the verification of our control group with the general population are shown in Table 4. The mean values of the monitored somatic parameters and their ontogenetic development for both the volleyball players and the control group are presented in Fig. 1. The analysis of the differences between the means of the monitored parameters amid the individual age groups of the volleyball players and the control group is presented in Table 5. The results in Table 1 clearly imply that there are no differences in the individual age categories of the volleyball players and the control group girls with respect to age. No statistically or practically significant differences were found. Therefore, the differences found in the monitored somatic parameters are not the result of a different age of the girls. Considering the fact that all the monitored parameters had normal distribution, it was possible to use the parametric $t$-test for the comparison. The values of Cohen's d, characterising the practical significance of the differences between the individual age group's ontogenetic development of the monitored variables are presented in Table 5.

The results indicate that the basic anthropometric attributes of our control group show normal development with respect to the mean values of the Czech general population. The Ni values found ranged from −0.04 to +0.25 SD. The range of ±0.75 SD is considered to be the mean development of an attribute. Therefore, we can consider our control group to be general population.

No statistically or practically significant differences between VG1 and CG1 were found in the age category of G1 (12 years old). The value of Cohen's $d$ was lower than 0.5 in all cases. In other age categories (G2-G4), no statistically or practically significant differences between BM ($d < 0.5$) were found between the groups of volleyball players and the control groups in the corresponding age categories. Statistically significant differences were found in all other parameters and their practical significance was also confirmed. The volleyball players had significantly higher FFM values than the control group girls of the same age,

**Table 4  Mean values of basic parameters 6th NAS and comparison with control group.**

| Age group | BH (cm) M ± SD (6thNAS) Ni | BM (kg) M ± SD (6thNAS) Ni | BMI (kg/m$^2$) M ± SD (6thNAS) Ni |
|---|---|---|---|
| G1 | 157.6 ± 7.3 −0.02 SD | 47.1 ± 9.1 0.01 SD | 18.9 ± 3.0 −0.03 SD |
| G2 | 162.0 ± 6.6 –0.05 SD | 51.3 ± 8.9 0.21 SD | 19.5 ± 2.9 0.12 SD |
| G3 | 166.9 ± 6.3 0.01 SD | 58.1 ± 7.9 0.05 SD | 20.8 ± 2.6 0.05 SD |
| G4 | 167.3 ± 6.3 −0.04 SD | 59.5 ± 8.4 0.22 SD | 21.2 ± 2.8 0.25 SD |

**Notes.**

BH, body height; BM, body mass; BMI, body mass index; M, mean; SD, standard deviation; Ni, normalization index.

even though no significant differences in BM were found. Between VG2 and CG2, we determined an intermediate difference ($d \geq 0.5$), and we determined a major difference ($d \geq 0.8$) in the older age categories. The higher representation of FFM also corresponds with the higher values of SMM, even when expressed as percentage of their ratio in total BM. The differences found were at the level of a major difference ($d \geq 0.8$), an intermediate difference ($d \geq 0.5$) was determined only when comparing SMM (kg) between VG2 and CG2. The higher FFM values in the volleyball players corresponded with significantly higher values of their TBW (statistically and practically). When comparing VG2 with CG2, we determined an intermediate difference ($d \geq 0.5$), and we determined a major difference ($d \geq 0.8$) in the other age categories. The BF ratio in the volleyball players was significantly lower than in the control group. In the values expressed in kilograms (BF kg), we determined an intermediate difference ($d \geq 0.5$), and in the percentage of the BF to BM ratio, we determined a major difference ($d \geq 0.8$). Also, the VFA values were significantly lower in the volleyball players. The difference found was at the level of an intermediate difference ($d \geq 0.5$).

To assess the development and differences in the monitored somatic parameters in relation to the increasing chronological age in the volleyball players and in the control group, we used the mean values of the individual groups (Fig. 1). We analysed the differences related to the increasing age in the volleyball players separately from the control group girls. The practical significance was assessed using Cohen's $d$. Between the age of 12 and 13, there was a more considerable increase in BH and BM in the volleyball players ($d > 0.8$) than in the control group ($d = 0.5$). The increase in BM was manifested by a more considerable increase in FFM, SMM (kg) and TBW (l) ($d > 0.8$). The development of the monitored parameters does not change in the following years, both in the volleyball players and in the control group. Between the age of 13 and 16, only minor differences ($d < 0.5$) were determined in BF (%), VFA and SMM (%) in both the volleyball players and the control group, other parameters showed intermediate differences ($d \geq 0.5$). Between

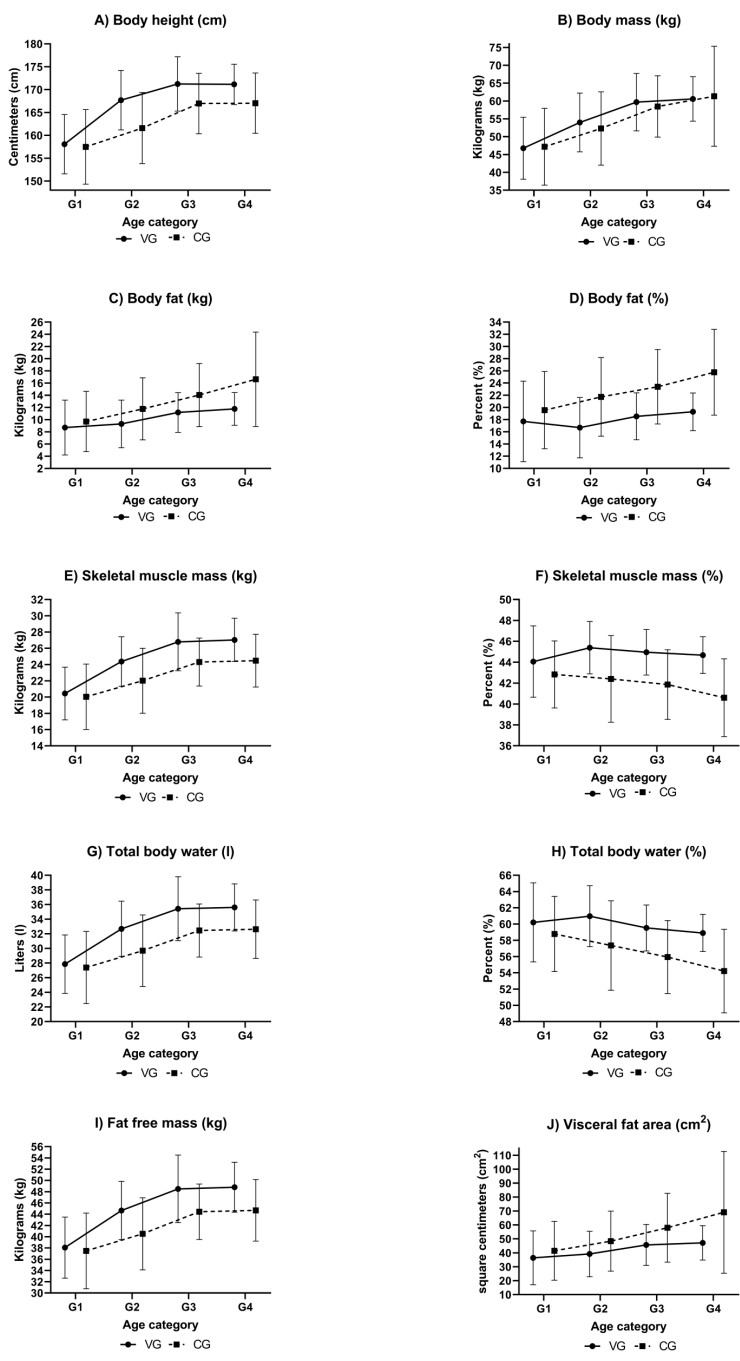

**Figure 1   Development of somatic parameters of female volleyball players and a female control group.**
(A) Development of body height (cm), (B) development of body mass (kg), (C) development of body fat
(kg), (D) development of body fat (%), (E) development of skeletal muscle mass (kg), (F) development of
skeletal muscle mass (%), (G) development of total body water (l), (H) development of total body water
(%), (I) development of fat free mass (kg), (J) development of visceral fat area (cm$^2$). Notes: VG, volley-
ball group; CG, control group.

**Table 5 Differences in somatic parameters between volleyball and control groups.**

| Parameters | VG1 vs. CG1 Difference | VG2 vs. CG2 Difference | VG3 vs. CG3 Difference | VG4 vs. CG4 Difference |
|---|---|---|---|---|
| BH (cm) | $0.59^{N+}$ | $6.03^{**++}$ | $4.28^{**++}$ | $4.11^{**++}$ |
| BM (kg) | $-0.41^{N+}$ | $1.68^{N+}$ | $1.22^{N+}$ | $-0.74^{N+}$ |
| TBW (l) | $0.46^{N+}$ | $2.99^{**++}$ | $2.98^{***+++}$ | $2.97^{***+++}$ |
| TBW (%) | $1.42^{N+}$ | $3.61^{***++}$ | $3.58^{***+++}$ | $4.69^{***+++}$ |
| BF (kg) | $-0.99^{N+}$ | $-2.47^{**++}$ | $-2.86^{**++}$ | $-4.86^{***++}$ |
| BF (%) | $-1.84^{N+}$ | $-5.06^{***+++}$ | $-4.87^{***+++}$ | $-6.48^{***+++}$ |
| VFA (cm$^2$) | $-5.03^{N+}$ | $-9.25^{*++}$ | $-12.32^{*++}$ | $-21.91^{**++}$ |
| FFM (kg) | $0.58^{N+}$ | $4.15^{**++}$ | $4.07^{***+++}$ | $4.11^{***+++}$ |
| SMM (kg) | $0.39^{N+}$ | $2.36^{**++}$ | $2.48^{***+++}$ | $2.55^{***+++}$ |
| SMM$_p$ (%) | $1.24^{N+}$ | $2.99^{***+++}$ | $3.09^{***+++}$ | $4.08^{***+++}$ |

Notes.

BH, body height; BM, body mass; TBW, total body water; BF, body fat; VFA, visceral fat area; FFM, fat free mass; SMM, skeletal muscle mass; SMMp, percentage of SMM in BM; VG, volleyball group; CG, control group; M, mean; SD, standard deviation; N, no significant.

$^*p < 0.05$.
$^{**}p < 0.01$.
$^{***}p < 0.001$.
$^+d < 0.5$.
$^{++}d \geq 0.5$.
$^{+++}d \geq 0.8$.

the age of 16 and 18, there were no significant differences in the volleyball players and in the control group, only minor differences ($d < 0.5$) were determined. It appears that the developmental tendencies do not differ, they only move towards better values in the volleyball players. It is also documented by the diagram of the development of somatic parameters in Fig. 1.

## DISCUSSION

To assess the physical activity represented by the participation of the volleyball players in training sessions and matches (Table 2), we used the recommendations stated in expert studies.. A daily physical activity of vigorous or moderate intensity that lasts 60 min is recommended for children and youth, which represents 420 minutes/week (*Janssen, 2007*; *Troiano et al., 2008*; *Janssen & Leblanc, 2010*; *Riebe et al., 2015*). This volume is not met in the youngest category (VG1) in the age categories we monitored, which has a physical activity of 360 minutes/week. This may be one of the possible reasons there was no difference in the monitored parameters between the volleyball players and the control group at the age of 12. Another cause might be the larger volume of spontaneous physical activity in the youngest girls of the general population, by which the girls make up for the absence of organised physical activities. Spontaneous physical activity decreases considerably as the age increases (*Bunc, 2018*). The decrease is then replaced with volleyball training and matches in older age groups. From the age of 13 (VG2), the volleyball players not only meet the recommendation of 420 minutes/week, but they also exceed it. It had an effect on the statistically and practically significant differences in the monitored parameters. The statistically and practically significantly higher BH in the volleyball players is an exception

as it cannot be linked to the higher volume of physical activity of the volleyball players. The high increase in BH cannot be caused by different ontogenetic development. In both groups (VG and CG), the girls are in the same chronological age and according to the expert studies are at the end of the peak height velocity (PHV). PHV is stated in the same period for girls both engaged and not engaged in sports. PHV for girls engaged in sports occurs at the age of 11.8-12.3 and for those not engaged in sports it is 11.4-12.2. Later, PHV is only mentioned in gymnasts (*Malina & Geithner, 2011*; *Malina et al., 2015*). The higher BH of the volleyball players is related to the rules and the essence of volleyball, whilst also being an advantage for serving. Furthermore, previous research in tennis also indicates that a higher BH was advantageous in serving and the tennis players with a higher BH served with a higher velocity (*Vaverka & Cernosek, 2013*). The higher BH of the volleyball players is probably caused by the selection criteria of clubs focusing on girls with a higher BH. This is also confirmed by results from studies that focused on the selection of female players for the national junior volleyball team. The selected players had a significantly higher BH than the players who were not selected (*Papadopoulou et al., 2019*; *Tsoukos et al., 2019*). The BH results presented in scientific studies correspond with the values determined in our volleyball players from VG2 (aged 13+), unless the studies deal with players selected for representative purposes. The values in those studies range from $167.0 \pm 8.0$ cm to $169.0 \pm 6.0$ cm (*Nikolaidis et al., 2012*; *Papadopoulou et al., 2019*; *Papadopoulou et al., 2020*). The BH values in young female volleyball players from the representation selections already exceed 170 cm at the age of 13 (*Nikolaidis et al., 2017*; *Papadopoulou et al., 2019*). The volleyball players in our youngest age category VG1 (12 years) have lower BH values than the twelve-year-old players in the study by *Nikolaidis et al. (2012)* where the mean BH value stated for such girls is $161.5 \pm 8.0$ cm. The difference is probably due to the fact that the selection of players for A teams competing at the highest level are often moved to a higher age category with regard to the lower number of players in Czech volleyball clubs.

Considering the fact that no statistically or practically significant differences were found in the values of BM in the volleyball players and the control group, we can not only compare the percentage ratio of the individual tissues in the total BM, but also their absolute values in kilograms. The primarily measured parameter in the BIA method is water, therefore, it is also required to analyse organism hydration as other parameters are calculated additionally on the basis of the primary parameter values. The TBW values closely correspond with the volume of muscle mass, presented by the SMM parameter. SMM is a body tissue that produces work and is developed by regular training (*Malina, 2007*). This was also confirmed in the volleyball players we monitored whose SMM ratio was much higher from the age category of VG3 than the CG girls. The higher FFM and SMM to BM ratio in the volleyball players is also related to their considerably lower BF ratio. The differences found were higher not only than TEM used in our laboratory for measurement standardisation (Table 3), but also higher than inter-daily variability, which ranges from 0.7 to 1.3 kg and from 0.9 to 1.6% in relation to the used BIA analyser and the growing interval between the individual measurements (*Vicente-Rodríguez et al., 2012*; *Kutáč, 2015*). The significantly lower BF ratio and higher SMM ratio determined in our volleyball players at the age of 13 and above (VG2) when compared with CG is

the result of their regular athletic preparation and it is a condition for adequate athletic performance. Body composition, especially the BF and SMM ratios, is an important factor that influences the athletic performance (*Nikolaidis et al., 2017*; *Papadopoulou et al., 2019*; *Papadopoulou et al., 2020*). The effect of BF on athletic performance is described well by the correlation between the skinfold thickness and performance indexes (Abakov jump, hand grip muscle strength, physical working capacity, peak power, sit-and-reach test). The higher the skinfold thickness, the more negative its influence (*Papadopoulou et al., 2020*). The same negative effect of BF was also demonstrated in a study that focused on performance in the above-jump test in 13-year-old female volleyball players (*Nikolaidis et al., 2017*). When there was absence of a difference in BF, there was also absence of a difference in performance indexes (*Papadopoulou et al., 2019*). We can only compare the determined values of the soft tissue ratio (BF and SMM) with the results of a study that used the same BIA analyser as we did. Therefore, we compared our results with the results of a study by opi et al. (2014). Even though the study authors analysed elite adult female volleyball players, the differences in the BF results in our players were insignificant. The authors state the mean BF ratio value of 17.6 ± 2.4%. The values of our players range from 16.7 ± 4.9 to 19.3 ± 3.1%. The largest difference was found in our oldest players (VG4), whose mean BF value was 19.3 ± 3.1%. Our VG2 players (13 years old) were the closest to the values of elite adult female players in the SMM ratio, the mean value of which in the aforesaid study was 46.1%, which is related to the fact that this age group had the lowest BF ratio. We did not find any other similar data in the available scientific literature, as the values of our oldest players (VG 4), where it is possible to assume that the ontogenetic development has ended and who have been trained for the longest period of time, should show values that are the closest to the published values of elite adult players. To assess the health condition of an individual, however, the ratio of subcutaneous fat and visceral fat, need to be monitored as they are more active metabolically and its increase is considered to be a risk factor not only in obesity, but also in cardiovascular diseases (*Beaufrère & Morio, 2000*; *Van Gaal, Mertens & De Block, 2006*; *Haberka et al., 2018*). In this study, visceral fat is expressed as an area (VFA). From the age of 13, the volleyball players had significantly lower values of this fat than the control group girls. When compared with the CG values, the lower BF ratio and the lower VFA values in our volleyball players are very positive for their state of health, especially considering the increasing prevalence of obesity in the child population (*Ng et al., 2014*). Considering that child obesity is carried over into adulthood where it may potentially increase morbidity and thus, impair the quality of life (*Tsigos et al., 2008*), we may consider regular volleyball training to be a suitable activity for maintaining reasonable body mass, including the body fat values.

No differences were found in the comparison of the development of the monitored parameters in the volleyball players and the control group girls in relation to the increasing chronological age. The values of somatic parameters gradually increase with the increasing chronological age up to the age of 16 in both groups (VG, CG), after that the development slows down and the differences are insignificant. The development of the monitored parameters corresponds with the course of development described in expert studies (*Malina & Geithner, 2011*). A considerable difference in the monitored parameters in the

**Table 6  Differences in the somatic parameter values at increasing chronological age expressed by Cohen's *d*.**

| Parameters | 12 vs. 13 years | | 13 vs. 16 years | | 16 vs. 18 years | |
|---|---|---|---|---|---|---|
| | VG | CG | VG | CG | VG | CG |
| BH (cm) | 1.5 | 0.5 | 0.6 | 0.7 | 0.0 | 0.0 |
| BM (kg) | 0.9 | 0.5 | 0.7 | 0.7 | 0.1 | 0.2 |
| BF (kg) | 0.1 | 0.4 | 0.5 | 0.5 | 0.2 | 0.3 |
| BF (%) | 0.2 | 0.3 | 0.4 | 0.3 | 0.2 | 0.4 |
| VFA (cm$^2$) | 0.2 | 0.3 | 0.4 | 0.4 | 0.1 | 0.3 |
| SMM (kg) | 1.3 | 0.5 | 0.7 | 0.7 | 0.1 | 0.1 |
| SMM$_p$ (%) | 0.4 | 0.1 | 0.2 | 0.1 | 0.1 | 0.4 |
| FFM (kg) | 1.2 | 0.5 | 0.7 | 0.7 | 0.1 | 0.0 |
| TBW (l) | 1.2 | 0.5 | 0.7 | 0.7 | 0.1 | 0.0 |
| TBW (%) | 0.2 | 0.3 | 0.4 | 0.3 | 0.3 | 0.4 |

Notes.

BH, body height; BM, body mass; BF, body fat; VFA, visceral fat area; SMM, skeletal muscle mass; FFM, fat free mass; TBW, total body water; VG, volleyball group; CG, control group.

volleyball players between the age of 12 and 13 is an exception. Significant increases that can be described as major differences ($d \geq 0.8$) were found in the parameters of BH, BM, SMM (kg) and TBW. The increased BH in the volleyball players between the age of 12 and 13 corresponds with the increase in BM. The considerable increase in BM is also accompanied by a considerable increase in the representation of the individual tissues (SMM, FFM, TBW), but it was not accompanied by a significant increase in body fat. The gradual significant increase in BF in our volleyball players probably does not occur thanks to the balanced energy intake and output. In practice, it means there is no increase in body fat due to a disturbed energy balance. The amount of body fat only changes within ontogenesis as a result of maturing of all the monitored girls. The energy output in girls doing sports is higher than in the girls at the same age without regular sports training, which increases the fitness of the girls doing sports (*Nikolaidis et al., 2012*).

The energy intake in the regular adolescent population without regular physical activity ranges from $8.76 \pm 2.36$ MJ/day to $9.28 \pm 2.0$ MJ/day (*Forrestal, 2011*; *Murakami & Livingstone, 2016*). The mean daily energy output in female volleyball players is $14.55 \pm 2.53$ MJ (*Woodruff & Meloche, 2012*), which is a value higher by 36.2–39.8% when compared with the values stated for individuals without regular physical activity. The daily intake of female volleyball players was $14.37 \pm 4.90$ MJ (*Woodruff & Meloche, 2012*), which confirms the energy balance. This energy balance is required to manage the load during training. It is thus obvious that volleyball training will not lead to reduced BM, but it could prevent BF from increasing. There is an increase in the energy intake in the regular population due to a lack of sufficient physical activity (*Vadiveloo, Zhu & Quatromoni, 2009*), which was demonstrated as permanent in our study. It is confirmed by the significant differences in BG between VG and CG in all age groups starting at VG2 and the absence of difference in BF values during ontogenesis (Table 6). Hypotheses H1 and H2 were confirmed.

## Limitations

There are several limitations to this study. The first one arises from the method used for the measurement of body composition. We used the bioelectrical impedance method (BIA) which is sensitive to the hydration of the organism. All the subjects (their legal guardians) were informed of the principles that needed to be observed before the measurement, however, it is not possible to ensure and verify the actual observance of all the principles for measurement in practice. The second limitation concerns monitoring the menstrual cycle phases that may have an effect on the resulting body composition values. With regard to the fact that we were not able to monitor the phases, only the occurrence of menstrual bleeding was an exclusion criterion in the measurement. The third limitation concerns the use of the values of basic anthropometric parameters (BH, BM, BMI) from 6th NAS, which was implemented in 2001. However, there have not been any updated normative values of the Czech population. The fourth limitation is the absence of checking the diet and energy balance of the monitored subjects. However, the volleyball players are not provided any common diet or a nutritionist within their training. The energy intake and output values are based on the values published in scientific studies.

## CONCLUSIONS

The study results show that regular volleyball training leads to a lower body fat ratio, lower visceral fat values and a higher skeletal muscle mass ratio compared with the general population. Therefore, it is possible to recommend regular volleyball training as a physical activity for maintaining adequate body mass.

The body fat ratio, the visceral fat and skeletal muscle mass values gradually increase with the increasing age of the volleyball players. However, those changes correspond with the changes in the general population. It was proven that the development of body composition parameters is subject to ontogenetic development having a higher effect than the volleyball training recorded.

Body fat is a frequently monitored parameter in training practice. Its gradual increase during ontogenesis is a natural feminine effect, as the results of our study showed. This fact should also be accepted by trainers in their preparation and management of the training process, especially in adolescent female athletes.

### Funding
The authors received no funding for this work.

### Competing Interests
The authors declare there are no competing interests.

## Author Contributions

- Petr Kutáč conceived and designed the experiments, performed the experiments, analyzed the data, prepared figures and/or tables, authored or reviewed drafts of the paper, and approved the final draft.
- David Zahradnik conceived and designed the experiments, authored or reviewed drafts of the paper, and approved the final draft.
- Miroslav Krajcigr analyzed the data, prepared figures and/or tables, and approved the final draft.
- Václav Bunc conceived and designed the experiments, prepared figures and/or tables, authored or reviewed drafts of the paper, and approved the final draft.

## Human Ethics

The following information was supplied relating to ethical approvals (i.e., approving body and any reference numbers):

The study was approved by the Ethical Committee of the Faculty of Education at the University of Ostrava (PdF OU č. 18/2018).

## Data Availability

The raw measurements are available in the Supplemental File.

## Supplemental Information

Supplemental information for this article can be found online at http://dx.doi.org/10.7717/peerj.9992#supplemental-information.

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
