# Peer review of "The effect of long-term volleyball training on the level of somatic parameters of female volleyball players in various age categories"

_PeerJ, doi:10.7717/peerj.9992_

## Round 0.1 · original submission · Major Revisions

Dear authors,

After reading the reviewers' reports, I think your work has scientific merit, but there are several issues that you must address in a revised version of the text. Please, see the comments below so as to have more information.

Best regards,
Dr Palazón-Bru

Reviewer 1 ·

Basic reporting

See below

Experimental design

See below

Validity of the findings

See below

Additional comments

General comment
This an interesting study on volleyball and the authors used correct scientific methods. Considering the popularity of this sport especially among female athletes, the findings have large practical applications. A novelty of this study - compared to previous studies - is the use of control group. I would recommend it for publication once the authors addressed a few concerns.

Specific comments
1. Text: Use paragraphs of 8-12 lines to improve the readability.
2. Abstract: Reduce the part of aims and methods.
3. Abstract: Increase the part of results adding more numbers, means, SD, p values, effect sizes.
4. Abstract: Revise the conclusions (l.30-33) to represent the specific findings of this study.
5. l.38-41: It is not necessary. Start directly with volleyball.
6. l.63-66: Add references.
7. l.87: Add 2-3 sentences with references presenting the rationale of this study, i.e. what is missing in the existed literature and why it is important to study it.
8. l.87-89: Revise the aim. The aim is whether age-related differences vary between volleyball and control group.
9. l.89: Add hypotheses and references.
10. l.92: When the study was conducted?
11. l.120: Where it took place?
12. l.223: The discussion should be revised totally. Use a first paragraph to summarize the findings, then start each paragraph with one of your findings, present if it agrees or differs with the literature and explain why.
13. l.295: Shorten the limitations. Revise this section to ‘limitations, strength and practical applications’ and add these aspects with an emphasis on practical applications.
14. l.316: Revise the conclusions so they are specific to the findings.
15. l.332: References: They should be revised in a large part. Many references are inappropriate (e.g. conferences, sites, books, journals with no IF) and they should be replaced by journals with high IF and published in the last 5 years.
16. l.332: Major literature is missing (e.g. doi: 10.1519/JSC.0b013e31823f8c06, 10.3390/medicina56040159, 10.3389/fpsyg.2019.02737, 10.23736/S0022-4707.16.06298-8).

Reviewer 2 ·

Basic reporting

1. Basic reporting
a. The abstract starts with a basic sentence that starts the entire motivation for this study: “Volleyball is often used as an activity for the cultivation of fitness and for influencing body composition (…)”. This statement is provided without any reference to support it. Furthermore, having been a volleyball coach for 20 years, I have never had an athlete engaging in volleyball practice with the goal of improving body composition. There are many potential reasons for a person to start playing volleyball, but improving body composition is likely but be one of the least common.
b. Overall, we do understand the rationale behind this study and it presents novelty and is – we believe – relevant to understand the real merits of practicing a given sport, even if only analyzing one dimension (in this case, body composition). However, the authors do not always choose the best line of thought. For example, work to rest ratios in volleyball clearly favor the pauses, i.e., for each second of work in the match, the athlete has 4 to 5 seconds of rest. Thus, perhaps in comparison with other team sports, volleyball is not the best example of demanding high levels of fitness and, therefore, might not be the best sport to promote changes in body composition.
c. In lines 50-52, the argument provided by the authors may even detract from their goal, because having a higher percentage of body fat may protect the joints when landing after a jump.
d. In lines 63-66, there are very strong statements made, but without any references to support them.
e. In lines 74-77, there is support for the need of engaging in physical activity…but the argument is made and sustained for children, while this study focused on teenagers.
f. In lines 84-85, again there are strong statements without any references to support the claims.
g. English language is clear throughout the document, with only a few minor mistakes that could benefit from a detailed review.
h. Globally, the theme seems relevant and novel, but is not always properly sustained and the lines of argumentation often do not support the authors’ intentions.
i. Structure, figures, raw data: authors fulfil PeerJ requirements.

Experimental design

2. Experimental design
a. Research design within scope of the journal. Research question was well defined, but not exactly well supported due to previously mentioned problems with the rationale.
b. Solid experimental design, and with a big sample. Both groups are well described, as well the experimental conditions. There is detailed information to fully understand the methods and possibly replicate them in the future.
c. One major flaw is that there was no reporting on weekly caloric intake. This will have a major impact on the findings, as I will describe later.
d. It is not clear at all how the estrogenic phase of the menstrual cycle was determined for each participant. If 28-day averages were used, they are very flawed, as natural interindividual variability in menstrual cycle can range from 23 to 35 days.
e. Another huge problem is that there is not reporting of the Typical Error of Measurement for the devices that were used (e.g., bioelectric impedance), nor is there any report of data reliability. This means that trustworthiness of data cannot be assessed and, therefore, casts a major shadow on data interpretation.
f. Otherwise, the statistical approach seems appropriate.
g. Overall, points c) and especially e) make it impossible to properly evaluate the actual results of this research.

Validity of the findings

3. Validity of the findings
a. The lack of information concerning weekly caloric intake makes it very difficult to conclude anything about the effects of volleyball in body composition, especially in body fat. In fact, by engaging in a weekly training volume that is way superior to that of the control group, the experimental group will likely have a much greater caloric need. This may be reflected on nutritional habits. Hence, volleyball may diminish body fat, but this effect may be hindered by increased caloric intake to face the demands of a strict training regimen. This fact should be widely recognized by the authors across the manuscript.
b. Findings cannot be properly interpreted without consideration of factor a), plus there is not information on TEM or ICCs. Therefore, I cannot assess if the findings are valid, because insufficient information appertaining the quality of data has not been provided.
c. Therefore, conclusions cannot be properly assessed in the current form of this manuscript.

Additional comments

4. General comments
a. Interesting premise, but the rationale has to improve in terms of discourse and referencing.
b. Crucial information is lacking in the methods sections, compromising any analysis of results and discussion.

Reviewer 3 ·

Basic reporting

The introduction is easy to read. However, in my opinion, it is necessary to extend existing knowledge on relationship concerning performance and sport practice-related differences. The introduction should include more update references regarding the association between sport-specific training background and physical performance. Moreover, I suggest you to support the following sentence “Regular and adequate training since childhood should not only lead to the development of fitness and motor performance, but also to the development of a series of personal traits of the individuals” including an update reference.

After the purpose statement, please provide a hypothesis for what the authors think the results will yield.

Experimental design

Some important information appears to be presently omitted from the methods section. Further description of the sampling procedure would be helpful for the reader. The recruitment process is a bit unclear. Please explain better how was selected the sample size and how was the data collected. When were the tests administered? Was the time of year, season, and time of day consistent for all subjects sampled? Further explanation about who collected the data is also necessary here. Some important information also appears to be presently omitted from the methods and results section. Have you tested the reliability of your data? If yes, please include the results.
It is not clear who conducted the assessments or whether they were blinded to group allocation.

Validity of the findings

In general, the first paragraph of the discussion should at least state which hypotheses were supported. Then the authors should follow with how their results compare with similar data, and what the authors results adds to the literature (different / unique aspects of the data). Several points are made in the discussion, but it is not clear to this reviewer how results from the current study are novel or add to the literature.
The authors did not discuss what is novel about this research or what it offers in terms of health implications. The authors did not discuss how this research may be disseminated into greater practice. Moreover, the limitations and the strengths of this research were not discussed at all.

Additional comments

While the topic is of interest in its current form it will require a lot of work before publication. The principal aim of this study was to assess the effect of volleyball training of girls at an older school age and adolescent girls on their somatic parameters.
The paper is interesting but there are many examples of poor sentence construction throughout. I accept that this is probably due to English not being the authors’ first language but it would need very careful editing and proof reading prior to publication.

---

## Round 0.2 · accepted · Accept

All the reviewers' concerns have been correctly addressed.

Reviewer 1 ·

Basic reporting

no comment

Experimental design

no comment

Validity of the findings

no comment

Additional comments

The authors addressed successfully all my concerns. I recommend its publication in its current form and believe that it will make a major contribution in its field.